# Impact of an In-Hospital Endocarditis Team and a State-Wide Endocarditis Network on Perioperative Outcomes

**DOI:** 10.3390/jcm10204734

**Published:** 2021-10-15

**Authors:** Mahmoud Diab, Marcus Franz, Stefan Hagel, Albrecht Guenther, Antonio Struve, Rita Musleh, Anika Penzel, Christoph Sponholz, Thomas Lehmann, Henning Kuehn, Karim Ibrahim, Marcus Jahnecke, Holger Sigusch, Henning Ebelt, Gloria Faerber, Otto W. Witte, Bettina Loeffler, Michael Bauer, Mathias W. Pletz, P. Christian Schulze, Torsten Doenst

**Affiliations:** 1Department of Cardiothoracic Surgery, Jena University Hospital—Friedrich Schiller University of Jena, 07743 Jena, Germany; antonio.struve@uni-jena.de (A.S.); gloria.faerber@med.uni-jena.de (G.F.); doenst@med.uni-jena.de (T.D.); 2Center for Sepsis Care and Control (CSCC), Jena University Hospital—Friedrich Schiller University of Jena, 07743 Jena, Germany; stefan.hagel@med.uni-jena.de (S.H.); bettina.loeffler@med.uni-jena.de (B.L.); michael.bauer@med.uni-jena.de (M.B.); mathias.pletz@med.uni-jena.de (M.W.P.); 3Department of Internal Medicine I, Jena University Hospital—Friedrich Schiller University of Jena, 07743 Jena, Germany; marcus.franz@med.uni-jena.de (M.F.); christian.schulze@med.uni-jena.de (P.C.S.); 4Institute for Infectious Diseases and Infection Control, Jena University Hospital—Friedrich Schiller University of Jena, 07743 Jena, Germany; 5Department of Neurology, Jena University Hospital—Friedrich Schiller University of Jena, 07743 Jena, Germany; albrecht.guenther@med.uni-jena.de (A.G.); rita.musleh@med.uni-jena.de (R.M.); otto.witte@med.uni-jena.de (O.W.W.); 6Institute of Medical Microbiology, Jena University Hospital—Friedrich Schiller University of Jena, 07743 Jena, Germany; anika.penzel@med.uni-jena.de; 7Department of Anaesthesiology and Critical Care Medicine, Jena University Hospital—Friedrich Schiller University of Jena, 07743 Jena, Germany; christoph.sponholz@med.uni-jena.de; 8Center of Clinical Studies, Jena University Hospital—Friedrich Schiller University of Jena, 07743 Jena, Germany; thomas.lehmann@med.uni-jena.de; 9Thuringia Clinics Georgius Agricola Saalfeld, Clinic for Internal Medicine III, 07318 Saalfeld, Germany; hkuehn@thueringen-kliniken.de; 10Chemnitz Clinic, Internal Medicine I, 09113 Chemnitz, Germany; karim.ibrahim@skc.de; 11St. Georg Clinic Eisenach, Clinic for Internal Medicine I, 99817 Eisenach, Germany; jahnecke.marcus@stgeorgklinikum.de; 12Heinrich-Braun-Clinic Zwickau, Clinic for Internal Medicine I, 08060 Zwickau, Germany; holger.sigusch@hbk-zwickau.de; 13Catholic Hospital St. Johann Nepomuk Erfurt, Clinic for Internal Medicine II, 99097 Erfurt, Germany; henning.ebelt@kkh-erfurt.de

**Keywords:** infective endocarditis, endocarditis team, endocarditis network

## Abstract

Background: Infective endocarditis (IE) requires multidisciplinary management. We established an endocarditis team within our hospital in 2011 and a state-wide endocarditis network with referring hospitals in 2015. We aimed to investigate their impact on perioperative outcomes. Methods: We retrospectively analyzed data from patients operated on for IE in our center between 01/2007 and 03/2018. To investigate the impact of the endocarditis network on referral latency and pre-operative complications we divided patients into two eras: before (*n* = 409) and after (*n* = 221) 01/2015. To investigate the impact of the endocarditis team on post-operative outcomes we conducted multivariate binary logistic regression analyses for the whole population. Kaplan–Meier estimates of 5-year survival were reported. Results: In the second era, after establishing the endocarditis network, the median time from symptoms to referral was halved (7 days (interquartile range: 2–19) vs. 15 days (interquartile range: 6–35)), and pre-operative endocarditis-related complications were reduced, i.e., stroke (14% vs. 27%, *p* < 0.001), heart failure (45% vs. 69%, *p* < 0.001), cardiac abscesses (24% vs. 34%, *p* = 0.018), and acute requirement of hemodialysis (8% vs. 14%, *p* = 0.026). In both eras, a lack of recommendations from the endocarditis team was an independent predictor for in-hospital mortality (adjusted odds ratio: 2.12, 95% CI: 1.27–3.53, *p* = 0.004) and post-operative stroke (adjusted odds ratio: 2.23, 95% CI: 1.12–4.39, *p* = 0.02), and was associated with worse 5-year survival (59% vs. 40%, log-rank < 0.001). Conclusion: The establishment of an endocarditis network led to the earlier referral of patients with fewer pre-operative endocarditis-related complications. Adhering to endocarditis team recommendations was an independent predictor for lower post-operative stroke and in-hospital mortality, and was associated with better 5-year survival.

## 1. Introduction

Infective endocarditis (IE) is a serious disease, carrying a considerable risk of 1-year mortality [1,2]. Mortality can reach up to 50% in cases of prosthetic IE associated with renal failure [3], and 64% in cases with severe heart failure [4]. Early diagnosis, appropriate antibiotic therapy, and early surgery, when indicated, are essential to improve patients’ outcomes [5,6]. Cardiac surgery is required in more than half of IE patients and is usually indicated in advanced stages of the disease (i.e., ongoing infection despite appropriate antimicrobial treatment or the presence of complications such as heart failure or systemic embolism) [5,6]. Delaying surgery is accompanied by an increased risk of local disease progression through the invasion of cardiac tissues, development of sepsis or septic shock with multiple organ dysfunction (MOF), and embolic events [7]. Current guidelines recommend establishing an endocarditis team consisting of cardiologists, cardiac surgeons, infectious disease (ID) physicians, microbiologists, neurologists, neurosurgeons, imaging specialists, and others [5,6]. While some studies have already demonstrated the value of implementing a multidisciplinary endocarditis team in improving patients’ outcomes [8,9], there is little existing literature on the impact of a regional endocarditis network on decreasing diagnostic and referral latency and reducing pre-operative endocarditis-related complications.

## 2. Methods

### 2.1. The Old Protocol

In cases admitted to our hospital with symptoms suggestive of IE, samples for blood cultures were obtained and empirical antimicrobial treatment was started immediately, which was then modified after receiving the microbiological results of blood cultures. The physician in charge determined the antimicrobial therapy according to current guidelines. TEE was then performed, and in the presence of surgical indications cardiac surgeons were consulted in order to discuss treatment strategies. Patients with IE diagnosed in other hospitals were only referred to our hospital if they had indications for surgery according to the current guidelines or if they needed intensive care therapy. In patients with operative indications and pre-operative neurological complications, neurologists were consulted to determine the timing of surgery. After surgery, the type and duration of antibiotic therapy were determined by cardiac surgeons and/or anesthetists, also according to current guidelines.

### 2.2. Endocarditis Team Management

In 2011 we established a dedicated endocarditis team of cardiologists, cardiac surgeons, infectious disease (ID) physicians, anesthesiologists, and neurologists. For patients admitted with suspicion of IE, the ID physicians were consulted within 6 h. Samples for blood cultures were obtained and empirical antimicrobial treatment was started immediately, which was then modified after receiving the microbiological results of blood cultures. The ID specialist determined an individualized antimicrobial treatment for each patient based on the microbiological results, type of IE, and presence of organ dysfunction. TEE was then performed followed by consultation of the endocarditis team, which proofed the indications for surgery, discussed operative risk, and determined the timing of surgery. After surgery, the ID specialist determined the type and duration of antimicrobial treatment for each patient. Management by the endocarditis team increased progressively, starting with 15% of the surgically treated patients in 2011 and reaching 94% of them in 2018, as shown in Figure 1.

### 2.3. Establishment of a State-Wide Endocarditis Network

In 2015 we started to expand the activities of the endocarditis team beyond our hospital to include other referring centers and cardiologists. We organized symposia and meetings with referring cardiologists and family doctors to increase the awareness of IE and the importance of early diagnosis and management. We published an article about IE and the role of the endocarditis team in a local health journal, which reaches every physician in the federal state of Thuringia. We established an ID telephone consultation service for referring hospitals, available 24 h/7 d. Through this telemedicine consultation, patients with indications for referral in our hospital were determined. The indications for referral included: presence of surgical indications, prosthetic IE or implantable device IE, *S. aureus* or fungal IE, presence of IE-related complications or organ dysfunction, or cases with an uncertain diagnosis which needed further investigations, such as nuclear imaging. Patients without indications for referral in our endocarditis center were offered individualized antimicrobial treatment recommendations through our ID physicians. A cluster-randomized clinical trial comparing the utility of telephone ID consultations with the standard of care in patients with *S. aureus* bacteremia was performed between August 2015 and January 2019 in 21 non-academic hospitals in the federal state of Thuringia. These telephone ID consultations allowed for the earlier detection of patients with *S. aureus* IE. This trial strengthened the cooperation between our endocarditis center and the referring physicians.

### 2.4. Data Collection

Data included patient demographics and comorbidities, microbiological data, perioperative data (cardiopulmonary bypass time (CPB), cross-clamp time, and concomitant procedures), and relevant post-operative outcomes. Long-term follow-up was obtained by reviewing hospital medical records and interviewing patients or their physicians. The follow-up time for survival was measured from the date of operation to the date of death or the date of the last contact with the patient (date of discharge or date of last follow-up).

### 2.5. Outcome Definitions

The primary endpoints of this study were the time from the first symptom to referral and to surgery, the incidence of cardiac abscesses, pre-operative heart failure (defined as NYHA ≥ III), and pre-operative acute renal failure requiring hemodialysis, and the incidence of pre-operative stroke.

Secondary endpoints were in-hospital and 1-year mortality, new-onset post-operative stroke defined as progression of an existing pre-operative stroke or occurrence of a new-onset post-operative stroke, new-onset post-operative hemodialysis, duration of intensive care unit and hospital stay, and 5-year survival.

### 2.6. Statistical Analysis

To investigate the impact of the endocarditis network on referral latency and pre-operative complications we divided patients into two eras: before (*n* = 409) and after (*n* = 221) 01/2015. Continuous variables are expressed as mean ± standard deviation (SD) or median (interquartile range), according to their distribution. A two-sided Student’s *t*-test for independent samples was used to compare normally distributed continuous variables, and a two-sided Mann–Whitney U test was used for variables not normally distributed. Categorical variables are presented as absolute and relative frequencies, and a chi-square or Fisher’s exact test (two-sided) was used to compare groups regarding differences in proportions. To investigate the impact of the endocarditis team on post-operative outcomes, multivariate binary logistic regression models were fitted for in-hospital mortality and post-operative stroke with endocarditis team management, pre-operative artificial ventilation, mitral valve IE, and *S. aureus* IE as independent variables. For both models, odds ratios (ORs) with 95% confidence intervals (CIs) are provided for the risk factors. A logrank test was used to research differences in long-term mortality between the two groups, and Kaplan–Meier estimated the 5-year survival rates that are reported with a 95% CI for each group. All statistical analyses were performed using SPSS Statistics 22 software (IBM Corp., Armonk, NY, USA) as well as SAS 9.4 (SAS Institute, Cary, NC, USA).

## 3. Results

Figure 2 shows the flow diagram of our study population and the corresponding in-hospital mortality and stroke rates. Between 01/2007 and 03/2018 overall, 1121 patients with a diagnosis of IE were admitted to our center. Among them, 732 patients were admitted during the first era, between 01/2007 and 12/2014, and 389 patients during the second era, between 01/2015 and 03/2018. In-hospital mortality was similar in both eras with 22% vs. 19%, respectively. The incidence of stroke was lower in the second era compared to the first era (14% vs. 24%, *p* < 0.001). Surgery was performed on 56% and 57% of patients with IE during the first and second era, respectively. Only surgically treated patients (*n* = 630) were included in further analyses for this study. The total incidence of stroke (pre- plus post-operative) in surgically treated patients in the second era was almost half that in the first era (17% vs. 33%, *p* < 0.001).

Appendix A shows the percentage of patients diagnosed in our center or referred from other hospitals during the two eras. The percentage of patients referred from other hospitals during the second era was similar to that during the first era (89.5% vs. 86.5%, respectively, *p* = 0.165).

### 3.1. Pre-Operative Patient Characteristics

Table 1 shows pre-operative patient characteristics of the IE patients operated on during the first era between 2007–2014 compared to those operated on during the second era between 2015–2018. The demographic data were similar in both groups. The mode of acquisition of IE was nosocomial in 28% in the second era compared to 23% in the first era (*p* = 0.086). Vegetations ≥ 15 mm were more frequently found in patients during the second era compared to the first era, 166 (75%) vs. 118 (29%) (*p* < 0.001). During the second era, there were significantly more cases of isolated mitral valve IE (34% vs. 25) and fewer cases of isolated aortic valve IE (25% vs. 37). The incidence of multiple-valve (≥two valves) IE was similar in both eras (38% vs. 39%). In the first era, 24% of patients were managed according to the recommendations of the endocarditis team compared to 91% in the second era. The time between onset of symptoms to referral or to surgery during the second era was reduced to almost half that observed during the first era (7 days vs. 15 days and 9 days vs. 17 days, respectively, *p* < 0.001). Patients operated on during the second era had lower incidences of pre-operative endocarditis-related complications, such as severe congestive heart failure NYHA ≥ III (45% vs. 69%, *p* < 0.001), acute renal insufficiency requiring hemodialysis (8% vs. 14%, *p* = 0.026), pre-operative stroke (14% vs. 27%, *p* < 0.001), and a lower incidence of cardiac abscess (24% vs. 34%, *p* = 0.018).

Appendix A shows the microbiological findings of the IE patients operated on during the first era compared to those operated on during the second era. *S. aureus* was the causative organism in 34% of patients in the second era compared to 25% (*p* = 0.026) in the first era, while Streptococcus IE was more common in the first era (24%) compared to (15%) the second era (*p* = 0.005).

### 3.2. Operative Procedures and Operative Outcomes

Table 2 shows operative procedures and operative outcomes of IE patients operated on during the first era, between 2007–2014, compared to those operated on during the second era, between 2015–2018. There were more single-mitral-valve procedures (34% vs. 25%) and fewer single-aortic-valve procedures (25% vs. 37%) during the second era compared to the first one. The incidence of double- and multiple-valve procedures were similar in both groups. The intervalvular fibrous body had to be reconstructed (UFO operation) in 1% of patients during the second era compared to 3% during the first era. Concomitant coronary artery bypass grafting (CABG) was less frequently performed during the second period (9% vs. 17%).

During the second era, the incidence of new post-operative stroke was reduced to almost half its value during the first era (5% vs. 9%, *p* = 0.040).

In-hospital mortality and 1-year mortality were similar between the second and the first era (24% vs. 26% and 36% vs. 37%, respectively).

### 3.3. Management According to Endocarditis Team Recommendations

Patients treated according to the recommendations of the endocarditis team had a lower incidence of post-operative stroke (4% vs. 11%, *p* = 0.004), less post-operative hemodialysis (17% vs. 20%, *p* = 0.305), lower in-hospital mortality (18% vs. 32%, *p* < 0.001), and lower 1-year mortality (29% vs. 44%, *p* < 0.001).

A lack of endocarditis team management recommendations was an independent predictor for in-hospital mortality (adjusted OR: 2.12, 95% CI: 1.27–3.53, *p* = 0.004) and post-operative stroke (adjusted OR: 2.23, 95% CI: 1.12–4.39, *p* = 0.02), as shown in Appendix A, respectively.

### 3.4. Effect of the Endocarditis Network

In order to further analyze the role of the endocarditis network separately from the combined effect of both the endocarditis team and network, we divided patients into two periods. The first period represents the time since initiating the endocarditis team and before establishing the endocarditis network (2011–2014). The second period represents the time after establishing the endocarditis network. Appendix A shows the pre-operative endocarditis-related complications as well as the post-operative outcomes for the patients treated during these two periods. The time between the onset of symptoms to referral or to surgery during the years 2015–2018 was significantly shorter than that during the years 2011–2014 (7 days vs. 14 days and 9 days vs. 18 days, respectively, *p* < 0.001). Patients treated between 2015 and 2018 presented with significantly lower incidences of severe congestive heart failure NYHA ≥ III (45% vs. 67%, *p* < 0.001) and pre-operative stroke (14% vs. 26%, *p* = 0.001) compared to those treated during 2011 and 2014. Although the incidence of *S. aureus* IE was significantly higher between 2015 and 2018 (34% vs. 24%, *p* = 0.022), cardiac abscesses occurred less frequently (24% vs. 33%, *p* = 0.062). Post-operative outcomes were not different between the two periods, except for less post-operative stroke in patients treated between 2015 and 2018 (5% vs. 13%, *p* = 0.001).

### 3.5. Five-Year Survival

Appendix A shows the Kaplan–Meier survival estimates of patients operated on during the first era (blue line) compared to those operated on in the second era (red line). The median follow-up time was 27 (1–77) months in the first era compared to 15 (0–44) months in the second era. The estimated 5-year survival for patients operated on during the first era was 47% (95% CI: 0.41–0.52) compared to 50% (95% CI: 0.37–0.62); however, the difference was not statistically significant (adjusted hazard ratio: 1.05, 95% CI: 0.80–1.35, log-rank = 0.724).

Appendix A shows Kaplan–Meier survival estimates of patients managed according to the endocarditis team recommendations (red line) compared to those who were not treated according to the endocarditis team recommendations (blue line). The median follow-up time was 27 (1–51) months in the endocarditis team group compared to 15 (1–73) months. The estimated 5-year survival for patients treated by the endocarditis team was significantly higher, 59% (95% CI: 0.51–0.66) compared to 40% (95% CI: 0.35–0.46); adjusted hazard ratio: 0.65, 95% CI: 0.51–0.84, logrank < 0.001.

## 4. Discussion

Our results demonstrate that the establishment of an endocarditis network has the potential to lead to the earlier diagnosis and referral of patients, possibly resulting in fewer pre-operative endocarditis-related complications, i.e., lower incidence of pre-operative stroke, severe congestive heart failure, cardiac abscesses, and acute renal failure requiring hemodialysis. We also found that creating an endocarditis team in our hospital resulted in significantly reducing in-hospital mortality and post-operative stroke in addition to improving 5-year survival. Thus, our results strongly support an interdisciplinary approach to this complex disease, both inside our own hospital as well as in referring centers.

Our findings of reduced in-hospital and 1-year mortality with multidisciplinary management (18% vs. 32% and 29% vs. 44%, respectively) are in accordance with other studies [10,11,12]. In a study including 292 patients with native valve IE, Chirillo et al. [11] investigated the impact of the implementation of a standardized protocol on outcome. They demonstrated a significant reduction in in-hospital mortality from 47% to 13% in surgically treated patients since the establishment of a multidisciplinary endocarditis team. Kaura et al. [13] demonstrated in a more recent study that the involvement of an endocarditis team was an independent predictor of 1-year survival in patients with IE who were managed medically. Ruch et al. [10] analyzed data from 391 patients with IE. They found a non-significant decrease in in-hospital mortality (20% vs. 15%) after establishment of an endocarditis team [10]. In our study, we were able to notice the reduction in in-hospital mortality by comparing patients treated according to the recommendation of the endocarditis team to those not treated according to the endocarditis team recommendations over the whole study period (18% vs. 32%, *p* < 0.001). When comparing the two eras before and after 2015, there was no significant reduction in in-hospital mortality (24% vs. 26%). This finding may be explained by the dramatic change in the characteristics of IE between the second and the first era. In the second era, there was significantly more *S. aureus* IE (34% vs. 25%). *S. aureus* infection has been previously shown to be an independent predictor of in-hospital mortality (OR: 2.1 (95% CI: 1.3–4.1) [14]. In addition, there was more mitral valve IE (34% vs. 25%) and more frequently the finding of large vegetations ≥ 15 mm (75% vs. 29%) in the second era compared to the first one. Both findings have been previously shown to affect in-hospital mortality [14,15,16]. In addition, 24% of patients operated on in our center during the first era were already treated according to the endocarditis team recommendations, which had a positive effect on in-hospital mortality during that period. We set 01/2015 as a cut-off point between the two eras because this was the date of the initiation of our endocarditis network with referring physicians and hospitals. It is difficult to attribute the observed positive effects to either the endocarditis team or the endocarditis network. In an attempt to illustrate the role of the endocarditis network separately from that of the endocarditis team, we additionally compared patients treated between 2011 and 2014 to those treated between 2015 and 2018. We found that after establishing the endocarditis network, the times needed to diagnosis and consequently to surgery were dramatically reduced. In addition, endocarditis-related complications were also significantly reduced. These findings underscore the additive value of the endocarditis network to the well-recognized value of the endocarditis team. Thus, the effect of the endocarditis network was most clearly visible by the reduced diagnostic latency and referral time as well as by the improved pre-operative conditions of the referred patients. At the same time, the endocarditis team worked to improve the quality of management of patients with IE inside our hospital.

This information concerning the impact of an endocarditis network is new. Previous studies only provided indirect evidence. For instance, Buchholtz et al. found that ID consultation, either on the bedside or by telemedicine, was associated with a reduced incidence of acute renal failure [17].

A multidisciplinary endocarditis team improves the outcomes of patients with IE that reach a tertiary reference center [10,11,12]. Communications inside the multidisciplinary team are very crucial. In our point of view, regular weekly meetings, similar to meetings of heart teams, are not suitable for an endocarditis team due to the urgency of the disease. In stable patients admitted to hospitals with suspicion of IE, ID specialists should be consulted within few hours of admission, followed by contacting the dedicated cardiac surgeons, cardiologists, neurologists, and anesthetists to make a treatment plan for the patient. In unstable patients, these communications should occur immediately upon admission. Further meetings during ICU and hospital stays should follow to discuss further treatment strategies and new findings (e.g., echocardiography, brain imaging, blood culture, etc.). In order to maintain and even improve the high quality of an endocarditis team, there has to be regular analyses and discussions of patients’ data. Furthermore, exchanging the results of these analyses with other teams from other centers helps to share experiences gathered on such a rare but devastating disease.

Endocarditis networks increase the awareness of referring physicians and lead to the earlier detection and consequently earlier management of patients with IE. Local scientific annual meetings on IE should be encouraged. There should be a continuous exchange of patients’ data between reference centers and referring physicians. Telemedicine has grown exponentially in the era of the COVID-19 pandemic [18]. To date, the application of telemedicine in the management of IE is very limited. Telemedicine allows referring physicians to interact with reference centers and facilitates the diagnosis and decision making of difficult cases.

### Limitations of the Study

Our study is limited by its retrospective nature. As a tertiary center, there may be a referral bias leading to more inclusion of patients with surgical indications. There might be potential bias related to different treatment strategies over the years and related to a lack of documentation of the frequency of telemedicine consultations of our multidisciplinary team.

## 5. Conclusions

The establishment of an endocarditis network for referring hospitals led to the earlier referral of patients with IE, which possibly resulted in fewer endocarditis-related complications on admission (i.e., congestive heart failure, cardiac abscesses, pre-operative stroke, and pre-operative acute renal failure). A lack of endocarditis team management recommendations was an independent predictor for post-operative stroke, in-hospital mortality, and was associated with worse 5-year survival. Thus, our results strongly support an interdisciplinary approach to this complex disease, both inside our own hospital as well as with referring centers.

## Figures and Tables

**Figure 1 jcm-10-04734-f001:**
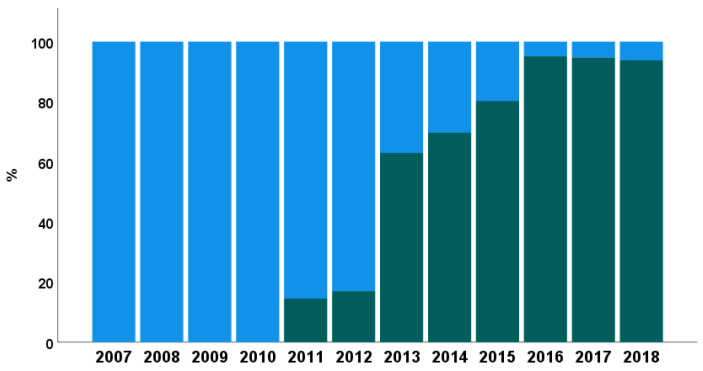
Bar chart showing the percentage of patients treated according to the recommendations of the endocarditis team over the years, from 2007 to 2018. Blue: patients not managed according to endocarditis team recommendations. Green: patients managed according to endocarditis team recommendations.

**Figure 2 jcm-10-04734-f002:**
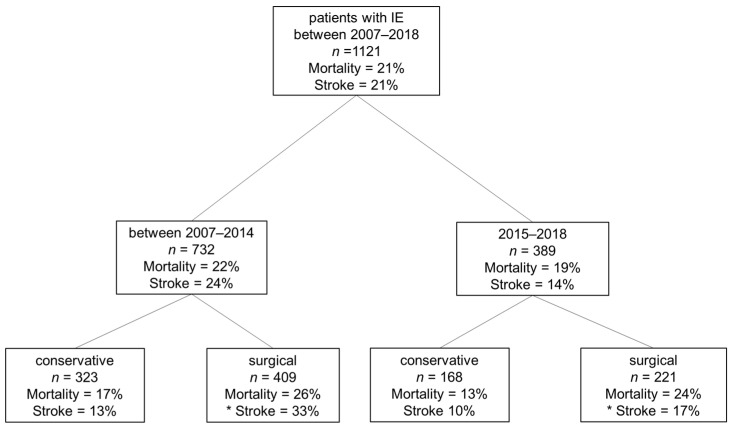
Flowchart of all patients with infective endocarditis treated in our center between 01/2007 and 03/2018 and the incidence of in-hospital mortality and stroke. IE: infective endocarditis; * Stroke: pre-operative and post-operative.

**Table 1 jcm-10-04734-t001:** Pre-operative patient characteristics for patients operated on in the first era (2007–2014) and in the second era (2015–2018).

	2007–2014 (*n* = 409)	2015–2018 (*n* = 221)	*p*-Value
Age (years)	65 (53–73)	67 (58–74)	0.086
Male sex	295 (72)	167 (76)	0.396
BMI	26.3 (24–30)	26.0 (24–30)	0.513
LVEF (%)	55 (50–60)	55 (46.5–60)	0.502
Insulin-dependent diabetes	78 (19)	33 (15)	0.228
Hyperlipidemia	167 (41)	89 (40)	0.932
COPD	33 (8.1)	22 (10.0)	0.460
Peripheral arterial disease	43 (11)	19 (9)	0.486
Poor mobility	78 (19)	38 (17)	0.592
CAD	119 (30)	54 (37)	0.147
Antiplatelet therapy	143 (35)	89 (40)	0.195
Cumarin therapy	49 (12)	37 (17)	0.063
Previous cardiac surgery	96 (24)	66 (30)	0.086
EuroSCORE II	16.5 (6.6–36.8)	16.3 (8.4–33.3)	0.987
Mechanical ventilation	73 (18)	44 (20)	0.522
Bilirubin	12 (8–21)	12 (8–23)	0.499
Creatinine	101 (78–155)	87 (69–130.5)	0.001
Hemodialysis dependency	91 (22%)	36 (16%)	0.078
Indication to surgery			<0.001
Heart failure	219 (54)	83 (38)	
Embolic risk	62 (15)	30 (14)	
Uncontrolled infection	128 (31)	108 (49)	
Nosocomial IE	92 (23)	59 (28)	0.086
Vegetation ≥ 1.5 cm	118 (29)	166 (75)	<0.001
Prosthetic valve IE	94 (24)	63 (29)	0.179
IE localization			<0.001
Mitral	103 (25)	74 (34)	
Aortic	150 (37)	56 (25)	
More or equal to two valves	156 (38)	85 (39)	
Management according to ET	98 (24)	200 (91)	<0.001
Time from symptoms to referral (days)	15 (6–35)	7 (2–19)	<0.001
Time from symptoms to operation (days)	17 (9–37)	9 (4–21)	<0.001
NYHA ≥ III	281 (69)	98 (45)	<0.001
Acute hemodialysis dependency	56 (14)	17 (8)	0.026
Pre-operative stroke	112 (27)	30 (14)	<0.001
Abscess	137 (34)	54 (24)	0.018

Data are given as a median (interquartile range, 25–75th percentile) or *n* (%). COPD: chronic obstructive pulmonary disease; CPB: cardiopulmonary bypass; IE: infective endocarditis; LVEF: left ventricular ejection fraction; BMI: body mass index; ICH: intracranial hemorrhage; CCS: Canadian Cardiovascular Society; ET: endocarditis team; NYHA: New York Heart Association.

**Table 2 jcm-10-04734-t002:** Post-operative outcomes for patients operated in the first era (2007–2014) and in the second era (2015–2018).

	2007–2014 (*n* = 409)	2015–2018 (*n* = 221)	*p*-Value
- Operative procedure			0.039
- Aortic valve	150 (37)	56 (25)	
- Mitral valve	103 (25)	74 (34)	
- Double-valve	97 (24)	50 (23)	
- Multiple-valve	16 (4)	11 (5)	
- Bentall	29 (7)	21 (10)	
- Reconstruction of IVF	14 (3)	3 (1)	
- Concomitant CABG	69 (17)	18 (9)	0.005
CPB time (min)	120 (92–169)	113 (88–160)	0.119
Cross-clamp time (min)	79 (58–108)	74 (57–106)	0.087
ICU stay (days)	4 (1–10)	7 (4–14)	<0.001
Hospital stay (days)	17 (10–28)	20 (14–28)	0.029
New post-operative hemodialysis	69 (17)	48 (22)	0.163
New post-operative stroke	38 (9)	10 (5)	0.040
In-hospital mortality	105 (26)	54 (24)	0.768
One-year mortality	152 (37)	79 (36)	0.795
Re-endocarditis	43 (11)	21 (10)	0.783

Data are given as a median (interquartile range, 25–75th percentile) or *n* (%). IVF: intervalvular fibrous body; CABG: coronary artery bypass grafting; CPB: cardiopulmonary bypass; ICU: intensive care unit; ICH: intracranial hemorrhage.

## Data Availability

All data are incorporated into the article and its online Appendix A. Additional data will be made available upon request in adherence with transparency conventions in medical research and through requests to the corresponding author.

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
