# Peer review of "Impact of an In-Hospital Endocarditis Team and a State-Wide Endocarditis Network on Perioperative Outcomes"

_jcm, 2021, doi:10.3390/jcm10204734_

Round 1

Reviewer 1 Report

The study could be published, according to my opinion. However, the authors have mistaken the legends of figure 1 and 2. They should be the opposite ones.

Author Response

We would like to thank all reviewers for the constructive and helpful comments. We have taken care to revise the manuscript according to the critiques. We addressed below each comment point by point:

Reviewer #1 

Comment: The study could be published, according to my opinion. However, the authors have mistaken the legends of figure 1 and 2. They should be the opposite ones.

Response: We thank the Reviewer for the supporting comment and apologize for the mistake. We corrected it.

Reviewer 2 Report

This study evaluated the impact of in-hospital endocarditis team including out-side telemedical network on clinical outcomes in patients with infective endocarditis (IE) by comparing those finding with a previous era of non-team approach. They clearly found the clinical valuable of adhering the team approach is an independent predictor for major adverse events complicated with IE at both acute and chronic periods. The present study was well-written to provide us a crucial viewpoints as to the current concept of clinical significance of multidisciplinary team for patients with IE. To strengthen this article, a couple of minor points should be addressed mentioned below.

First, as mentioned in the study design, they compared clinical outcomes between the two era before and after the establishment of their IE team. In the era of IE team, they also developed in-hospital working to referring other hospitals using telemedicine in 2015. So, as sub-analysis of the study, they should evaluate the value of those developments for their findings (e.g. comparing clinical outcomes between 2011-2015 and 2105-2018).

Second, as to the incidence in the long-term follow-up, a prevalence of re-IE and also hospitalized heart failure should be evaluated in addition to all-cause death.

Third, in discussion, if possible, the reviewer wish the authors’ viewpoints to maintain good quality of multidisciplinary IE team and also the well-balanced network.

Author Response

We would like to thank all reviewers for the constructive and helpful comments. We have taken care to revise the manuscript according to the critiques. We addressed below each comment point by point:

Reviewer #2

Comment 1: First, as mentioned in the study design, they compared clinical outcomes between the two era before and after the establishment of their IE team. In the era of IE team, they also developed in-hospital working to referring other hospitals using telemedicine in 2015. So, as sub-analysis of the study, they should evaluate the value of those developments for their findings (e.g. comparing clinical outcomes between 2011-2015 and 2105-2018).

Response: We thank the reviewer for this suggestion. We conducted a comparison between 2011- 2014 and 2015-2018. We found that after establishing the endocarditis-network, the times needed to diagnosis and consequently to surgery were dramatically reduced. In addition, endocarditis-related complications were also significantly reduced. We added a Supplementary Table 4 showing the results of this comparison and we commented on this table in the results section in page 8 line 285-305.

“3.4 Effect of Endocarditis Network

In order to further analyze the role of the endocarditis-network separately from the combined effect of both the endocarditis-team and network, we divided patients into two periods. The 1st period represents the time since initiating the endocarditis-team and before establishing the endocarditis-network (2011-2014). The 2nd period represents the time after establishing the endocarditis-network. Supplementary Table 4 shows the pre-operative endocarditis-related complications as well as the post-operative outcome for the patients treated during these two periods. The times between the onset of symptoms to referral and  to surgery during the years 2015-2018 are significantly lower than that during the years 2011-2014 (7days vs 14 days and 9 days vs 18 days, respectively, p<0.001). Patients treated between 2015 and 2018 presented with significantly lower incidences of severe congestive heart failure NYHA ≥ III (45% vs 67%, p<0,001) and pre-operative stroke (14% vs 26%, p=0.001) compared to those treated during 2011 and 2014. Although the incidence of S. aureus IE was significantly higher between 2015 and 2018 (34% vs 24%, p= 0.022), cardiac abscesses occurred less frequently (24% vs 33%, p=0.062). Post-operative outcomes were not different between the two periods, except for less post-operative stroke in patients treated between 2015 and 2018 (5% vs 13%, p=0.001). “

Supplementary Table 4: Pre-operative endocarditis-related complications as well as post-operative outcomes for the patients treated between 2011 and 2014 compared to those treated between 2015 and 2018.

2011-2014 (n=234)

2015-18 (n=221)

p-value

Time form symptom to referral

14 (6-33)

7 (2-19)

<0.001

Time form symptom to surgery

18 (10-38)

9 (4-21)

<0.001

Pre-operative NYHA ≥ III

157 (67)

98 (45)

<0.001

Pre-operative acute renal failure

28 (12)

27 (8)

0.157

Cardiac Abscess

76 (33)

54 (24)

0.062

Pre-operative stroke

60 (26)

30 (14)

0.001

Post-operative stroke

30 (13)

10 (5)

0.002

Post-operative hemodialysis

45 (19)

48 (22)

0.561

Length of ICU stay

5 (2-13)

7 (4-14)

0.992

Length of hospital stay

18 (12-34)

20 (14-28)

0.580

In-hospital mortality

63 (27)

54 (24)

0.316

One-year mortality

93 (40)

79 (36)

0.386

Re-endocarditis

23 (10)

21(10)

1.00

Data is given as median (interquartile range, 25th–75th percentile) or n (%).

ICU: intensive care unit; NYHA: New York Heart Association.

We also commented on this finding in the discussion page 9 line 368-380:

“In an attempt to illustrate the role of the endocarditis-network separately from that of the endocarditis-team, we additionally compared patients treated between 2011 and 2014 to those treated between 2015 and 2018. We found that after establishing the endocarditis-network, the times needed to diagnosis and consequently to surgery were dramatically reduced. In addition, endocarditis-related complications were also significantly reduced. These findings underscore the additive value of the endocarditis-network to the well-recognized value of the endocarditis-team. Thus, the effect of the endocarditis-network was visible the best by the reducing the diagnostic latency and the referral time and by improving the pre-operative condition of the referred patients. While the endocarditis-team worked to improve the quality of management of patients with IE inside our hospital.”

Comment 2: Second, as to the incidence in the long-term follow-up, a prevalence of re-IE and also hospitalized heart failure should be evaluated in addition to all-cause death.

Response: We thank the reviewer for the comment. We analyzed the incidence of re-endocarditis during follow-up and found it similar in both groups (11% in 1st era vs 10% in the 2nd era, p=0.783). We added this information to Table 2 in the main manuscript. However, we cannot perform Kaplan-Meier analysis of event-free survival as we do not have information on the time point of the event (re-endocarditis).

Regarding hospitalization due to   heart failure, we unfortunately do not have any information on it.

Comment 3: Third, in discussion, if possible, the reviewer wish the authors’ viewpoints to maintain good quality of multidisciplinary IE team and also the well-balanced network.

Response: We thank the reviewer for the constructive comment. We added in the discussion viewpoints to maintain good quality of the endocarditis team and network.

“Multidisciplinary endocarditis-team improves the outcome of patients with IE reaching a tertiary reference center 12-14. Communications inside the multidisciplinary team are very crucial. In our point of view, regular weekly meetings, similar to heart-team meetings, are not suitable for endocarditis team due to the urgency of the disease. In stable patients admitted to the hospital with suspicion of IE, ID specialists should be consulted within few hours of admission followed by contact with the dedicated cardiac surgeons, cardiologists, neurologists, and anesthetists to make the treatment plan for the patient. In unstable patients, these communications should occur immediately upon admission. Further meetings during ICU and hospital stay should follow to discuss further treatment strategies and new findings (e.g. echocardiography, brain imaging, blood culture, etc.). In order to maintain and even improve high quality of endocarditis-team, there has to be regular analyses and discussions of patients’ data. Furthermore, exchanging the results of these analyses with other teams from other centers helps sharing experiences gathered on such a rare but devastating disease.  

Endocarditis-networks increase the awareness of referring physicians and leads to earlier detection and consequently earlier management of patients with IE. Local scientific annual meetings on IE should be encouraged. There should be continuous exchange of patients’ data between the reference centers and referring physicians. Telemedicine has grown exponentially in the era of COVID-19 pandemic 20. To date, the application of telemedicine in management of IE is very limited. Telemedicine allows the referring physicians to interact with reference centers and facilitates the diagnosis and decision making of difficult cases.”

We hope that this revised version now meets your expectations and thank you and all the reviewers for your efforts that improved this manuscript.